# Evaluating the Capability of *Epipremnum aureum* and Its Associated Phylloplane Microbiome to Capture Indoor Particulate Matter Bound Lead

**DOI:** 10.3390/plants14192956

**Published:** 2025-09-23

**Authors:** Diego G. Much, Anabel Saran, Luciano J. Merini, Jaco Vangronsveld, Sofie Thijs

**Affiliations:** 1Scientific Research Agency, CONICET, Santa Rosa CP6300, La Pampa, Argentina; lucianomerini@yahoo.com.ar; 2Department of Biology, Centre for Environmental Sciences, Hasselt University, BE3590 Diepenbeek, Belgium; jaco.vangronsveld@uhasselt.be (J.V.); sofie.thijs@uhasselt.be (S.T.); 3Department of Plant Physiology and Biophysics, Faculty of Biology and Biotechnology, Maria Skłodowska-Curie University, 20-400 Lublin, Poland

**Keywords:** *Epipremnum aureum*, pothos, indoor air pollution, phylloplane microbiome

## Abstract

In this study we evaluated over a 1-year period, the ability of *Epipremnum aureum* leaves to collect particulate matter (PM)-bound Pb from an indoor environment. Using Illumina MiSeq, we investigated the changes in the phylloplane microbiome connected with the accumulation of this pollutant. Plants were placed in a shooting room, where PM release from each shot was recorded, along with PM_2.5_ and PM_10_ sequestration and leaf element enrichment by ICP. Additionally, black carbon (BC) sequestration was determined, and SEM-EDX was performed on leaves after 12 months of exposure. Our results indicated that ambient air pollution shapes microbial leaf communities by affecting their diversity. At the order level, Pseudomonadales, along with Micrococcales, appeared (at a low relative abundance) after exposure to indoor PM-bound Pb air pollution. This study provides a unique comparison of *Epipremnum aureum* air filtration performance between a standard office environment and a firearm shooting range. The air filtration approach holds promise for reducing indoor air pollution, but more knowledge about the underlying mechanisms supporting genera capable of coping with airborne pollutants is still required.

## 1. Introduction

The report by the European Environment Agency [1] underscores the high environmental and social concerns associated with particulate matter (PM) air pollution, presenting many challenges in terms of management and mitigation. Inhalable PM, characterized by a diameter less than 10 μm, comprises a complex mixture of organic substances, inorganic salts, and trace elements [2]. Recent studies have highlighted the substantial indoor exposure to PM, potentially surpassing outdoor levels [3,4,5,6,7]. For instance, a study performed in Mexico City has shown that in many homes indoor concentrations of PM_2.5_ can be two to five times higher than outdoor levels, driven largely by cooking, cleaning, smoking, and occupancy behavior, with significant spatial variability depending on household location, ventilation, and proximity to outdoor sources [8]. Distinctly, recent research has shown that firing ranges represent a significant indoor source of particulate matter, particularly containing heavy metals such as lead (Pb), antimony (Sb), tin (Sn), and copper (Cu) [9].

Moreover, PM exposure has been linked to a spectrum of severe respiratory (including asthma, lung cancer, etc.) and cardiovascular diseases [10,11,12]. Lee et al. (2023) [13], reported a case of a 31-year-old worker who developed occupational asthma after just one month of exposure in an indoor air gun shooting range. Daily peak expiratory flow (PEF) measurements revealed significant differences between workdays and days off, confirming the diagnosis. The study emphasizes that exposure to indoor air pollutants, particularly lead, combined with individual predispositions such as atopy and allergic rhinitis, played a key role in the development of asthma. This case underscores the critical importance of implementing preventive measures and adequate ventilation in indoor shooting environments to safeguard workers’ respiratory health.

With the growing interest in nature-based solutions (NbS), indoor plants have emerged as a promising solution for purifying indoor air and mitigating the adverse health effects of air pollution [14,15]. Phylloremediation, which harnesses the capabilities of leaves and their associated microbiome to bioremediate hazardous air pollutants, offers a novel approach [16]. Indoor leaf accumulation of PM has been reported for several plant species, including *Ipomoea batatas* L., *Hedera helix* L. and *Epipremnum aureum* [17].

Earlier research has highlighted the role of the phylloplane microbiome in capturing and oxidizing air pollutants [18,19]. With the total leaf area estimated at more than 508 million km^2^, the phylloplane serves as one of the largest microhabitats on Earth [20]. This microhabitat hosts diverse communities of microorganisms, with bacteria being the most abundant group closely associated with plants [21]. The composition and diversity of the phylloplane communities are significantly influenced by plant species, as suggested by Vogel et al. (2020) [22]. However, Laforest-Lapointe et al. (2016) [23] and later Stevens et al. (2021) [24] reported that host species only accounted for a portion of the factors influencing the, composition of phylloplane bacterial community.

Proteobacteria (alpha, beta, and gamma proteobacteria), Bacteroidetes, Firmicutes, and Actinobacteria are among the common bacterial groups inhabiting the phylloplane of various plant species [25]. Nonetheless, the survival of these microorganisms hinges on their capacity to develop specific resistance mechanisms to confront the adverse environmental conditions prevailing in the phylloplane [16].

In light of these considerations, we have evaluated the potential of *Epipremnum aureum* (devil’s ivy or pothos) to effectively lower the levels of indoor PM-Pb, as well as the effects of air quality on the phylloplane microbiome. To our knowledge, this is the first study directly comparing plant performance in a conventional office environment versus a firearm shooting range, thereby emphasizing the novelty of testing plant-based air filtration under such contrasting indoor conditions.

## 2. Results

### 2.1. PM and BC Concentrations in Leaf Wash Suspensions

Deposition of BC, PM_2.5_ and PM_10_ on *E. aureum* leaves was monitored over a period of 12 months, starting from August 2021 until August 2022. As shown in Figure 1A, PM_10_ concentrations increased significantly after 3 months of exposure at both the RR and SR sites. Similarly, Figure 1B shows the accumulation of PM_2.5_, which also increased significantly at both sites after 3 months. However, while PM_10_ and PM_2.5_ levels continued to rise in the RR plants at 6 months, a decrease was observed in the SR plants. Finally, after 12 months, both PM_10_ and PM_2.5_ tend to decrease at both sites. We also observed that those plants located in the SR after 3 months accumulated almost the double amount of PM_10_ (5.26 × 10^−04^ ± 3.55 × 10^−05^ mg·cm^−2^) and PM_2.5_ (4.89 × 10^−04^ ± 1.03 × 10^−05^ mg·cm^−2^) compared to RR (2.61 × 10^−04^ ± 8.69 × 10^−05^ mg·cm^−2^ and 2.79 × 10^−04^ ± 3.54 × 10^−05^ mg·cm^−2^, respectively). This is consistent with the measured concentrations of PM dispersed with each gunshot (Appendix A). At month 3, RR and SR are statistically different in both, PM_10_ and PM_2.5_ (*p* < 0.05). However, after 6 and 12 months the amounts of PM_10_ and PM_2.5_ did not show significant differences between both sites (Appendix A).

Additionally, we quantified the BC load of the leaves (Figure 1C). The results indicate that plants in the SR at T12 exhibited the highest BC load (bcH) (1.94 × 10^06^ ± 8.81 × 10^05^ particles·mL^−1^) while plants in the RR at T12 exhibited an intermediate BC load (bcM) (7.26 × 10^05^ ± 1.83 × 10^05^ particles·mL^−1^), with a highly significant difference compared to the SR at that time point (*p* < 0.0001). Furthermore, as expected, plants from both sites at T0 had similar values of BC (*p* > 0.05), which are considered the lowest BC load (bcL) category (Appendix A).

### 2.2. Leaf Elemental Composition and Surface Particle Element Distribution

#### 2.2.1. Leaf Elemental Composition by ICP

After 12 months, only the concentrations of Ca, Cd, Mg, and Zn differed significantly (*p* < 0.05) between RR and SR plants, whereas the other elements showed similar patterns without significant differences between sites (Table 1). Additionally, significant differences (*p* < 0.05) were observed for Mn in RR plants at T12 and for P in SR plants at T3, compared to their concentrations at the previous sampling time. More detailed information about the statistical results for element concentrations are presented in Appendix A. Pb enrichment in *E. aureum* leaves was probably not detected by ICP, due to the limited recovery percentages obtained by the digestion method (61–70% for Pb) and the ICP detection limit for Pb of 0.05 mg kg^−1^.

#### 2.2.2. Leaf Surface Particle Element Distribution by SEM-EDX

In Figure 2, SEM images and EDX spectra are presented of *E. aureum* leaves after 12 months of exposure in the RR and SR environments. The locations marked by numbers in black followed by a cross (Figure 2A), correspond to regions with more electron-dense (bright) plaques on the leaf surface. The EDX spectra provide the weight percentage (%) of elements present within the analyzed area (plaques at the highlighted locations) (Figure 2B). This comparison illustrates the differences in PM-metal composition between the two environments. The leaves from the SR environment show PM-Pb and PM-N enrichment. While, leaves from the RR environment sow PM-K, PM-Al and PM-Mg. Elements like Ca, Cl, Si and Na were found enriching PM in leaves from both sites. No consistent pattern was observed in the distribution of PM on the leaves, regardless of the site or the PM composition.

### 2.3. Phylloplane Bacterial Diversity Assessment

Following denoising of all libraries, ASV feature sequences and ASV tables were merged, resulting in a total effective sequence volume of 283,052 reads with an average of 14,152.6 reads per sample. After chimera removal, the total amount of high-quality sequences obtained was 259,864 with a mean value of 12,993.2.

Although species richness index did not show a significant difference between sites at T12 (Figure 3A), diversity and relative abundance (Figure 3B) and distribution of species (Figure 3C) were influenced by the indoor environment condition (*p* < 0.05). After 12 months of exposure, diversity and abundance of *E. aureum* phylloplane microbiome of plants located at the SR was significantly lower. Furthermore, differences of Pielou’s evenness index (Figure 3C) between the two environments suggest an uneven distribution with high densities of only few species in SR at T12, influenced by the SR conditions.

On the other hand, no differences in Faith’s PD (Figure 3D) were found between the environments, suggesting no shifts in the evolutionary breadth of the phylloplane microbiome.

The statistical analysis of the sampled ASVs produced a detailed table outlining the specific composition of phylloplane bacteria at each taxonomic level for each sample. This table facilitated the calculation of taxonomic unit compositions within each level for all twenty samples. The taxonomic classification of the inferred ASVs via the SILVA database revealed a total of 6 phyla, 9 classes, 21 orders, 28 families, and 34 genera. At the phylum level, phylloplane bacteria with a relative abundance exceeding 1% included Actinobacteriota, Cyanobacteria, Firmicutes, and Proteobacteria. Actinobacteriota appeared as the dominant phylum, with a relative abundance surpassing 69%. The comparative analysis of variations in microbial abundances at the phylum level between RR and SR at T0 and T12 (Figure 4A) illustrates how the microbial microbiome structure evolves over time and under different environmental conditions, suggesting notable shifts in the microbial microbiome, at least partly reflecting the specific impacts of the SR environment. The relative abundance of Proteobacteria decreased after 12 months in samples from both sites. On the other hand, the principal coordinate analysis (PCoA) plot of the same samples (Figure 4B) illustrates the clustering and dispersion of microbial communities, revealing that SR samples at T12 exhibit less dispersion compared to other groups.

To determine any variability in the bacterial phyllo-microbiomes between the two locations at the order level, the orders present in each of the samples were analyzed (Figure 5). The order level Corynebacteriales had the highest abundance in each sample (>70%) at both T0 and T12. In RR samples, a notable decrease in Pseudomonadales was observed after 12 months. In SR samples, by contrast, Pseudomonadales, along with Micrococcales, persisted among the few low-abundance orders. Notably, there was a marked diminution in low-abundance orders at SR samples at T12, compared to their presence in RR at the same time, which aligns with the observed reduction of microbial diversity mentioned above.

It is worth mentioning that, if we analyze more deeply at the genus level, the genus with the highest relative abundance (>70%) in each sample was *Rhodococcus* (associated with the Corynebacteriales Order), which showed increased abundance in the SR site (Appendix A).

## 3. Discussion

PM deposition on *E. aureum* leaves increased after three months, regardless of plant location. This upward trend slowed down at 6 and 12 months (Figure 1A,B). Our findings are consistent with those of Yu et al. (2018) [26], who reported that net photosynthetic rate and stomatal conductance tend to decline over time under elevated PM_2.5_ exposure. BC accumulation on *E. aureum* leaves (Figure 1C) was similar at baseline (T0) for both sites, but by T12 plants at the SR site showed markedly higher BC loads than those at the RR site. To the best of our knowledge, only one study is also reporting BC concentrations in indoor shooting ranges during official national competitions. Bisht et al. (2013) [27] monitored indoor air quality in stadiums during the 19th Common Wealth Games (CWG) in Delhi (India). They found that BC showed a positive correlation with CO, an air pollutant resulting from incomplete combustion. However, the present study is the first to employ plants as bioindicators for BC released during shooting activities. Witters et al. (2020) [28] investigated the potential of using plants as bioindicators for monitoring indoor air pollution, specifically combustion-derived particles (CDPs). Plants were shown to effectively reflect the presence and concentration of CDPs in indoor environments, offering a simple and cost-effective approach for assessing indoor air quality.

Variation in leaf Ca, Cd, Zn and Mg concentrations reflect plant responses to indoor PM released during shooting (Table 1). No reports exist regarding changes in trace element concentrations of *E. aureum* leaves due to exposure to polluted air. However, a study by Shah et al. (2025) [29] documented the phytotoxic effects of cigarette smoke on *E. aureum*, noting alterations in morphological and physiological parameters, which underscores the sensitivity of this species to airborne contaminants. Gajbhiye et al. (2022) [30] reported the foliar uptake patterns of Pb, Cd, and Cu bound to PM in two evergreen tree species, *Senna siamea* and *Alstonia scholaris*. The findings highlighted that indoor plants can absorb and accumulate heavy metals from the surrounding air, emphasizing their potential as bioindicators of indoor air pollution.

SEM-EDX images (Figure 2) illustrate the differences in PM-metal composition between the two environments. Santunione et al. (2024) [31] used SEM/x-EDS to analyze PM trapped by four different plant species in an urban forest in Italy. They concluded that the micromorphology of the leaves plays an essential role in determining the ability of each species to capture particulate matter.

Despite no changes in species richness or the evolutionary breadth were found, the indoor environment significantly reduced the diversity and abundance of the *E. aureum* phylloplane microbiome after 12 months (Figure 3). Actinobacteriota appeared as the dominant phylum (Figure 4) regardless of time or location, which is consistent with the findings of other studies, for instance, lettuce foliage coming from different geographical and seasonal scales [32], six species of *Populus* spp. [33] or perennial biofuel crops [34]. At the order level Corynebacteriales dominated across all samples (Figure 5) and at the genus level, *Rhodococcus* exhibited the highest relative abundance (Appendix A). There was a marked diminution in low-abundance orders at SR samples at T12, compared to their presence in RR at the same time (Figure 5), which aligns with the observed reduction of microbial diversity mentioned above (Figure 3B). By contrast, Pseudomonadales, along with Micrococcales, persisted at T12 in SR samples suggesting they may resist this specific exposure. Pátek et al. (2021) [35] described that toxic metals and metalloids may cause detrimental effects on bacteria through several mechanisms.

Currently, no studies have specifically evaluated the leaf bacterial community of *E. aureum* in response to different air pollutants. However, there are investigations addressing related aspects, such as *E. aureum* ability to purify air. For instance, a NASA study showed that *E. aureum* can remove up to 73% of carcinogenic air pollutants in indoor environments, including formaldehyde, benzene, and trichloroethylene [36].

While this study provides novel insights into the use of *E. aureum* as a bioindicator of indoor PM-Pb, was limited to two specific indoor environments, which may not capture the full variability of air pollution exposure in other settings. Future research should include a broader range of indoor environments, longer monitoring periods, and multi-omics approaches to better understand the interactions between PM composition, heavy metals, and phylloplane microbial communities. Additionally, investigating the physiological responses of E. *aureum* to different pollutant types could improve its application as a bioindicator in diverse indoor contexts.

## 4. Materials and Methods

### 4.1. Sampling

Ten *E. aureum* plants were obtained from a commercial nursery on 16 August 2021 (Agropecuaria, Santa Rosa, Argentina) and acclimated to assay conditions. Non-exposed, cleaned leaves were collected initially (0 month). Subsequently, five plants were placed in an office environment as a reference room (RR), while the remaining five plants were positioned inside a firearm shooting room (SR) at the Scientific Investigation Agency (refer to Appendix A), where measurable concentrations of PM are dispersed with each shot (Appendix A). Plants were watered once a week to prevent soil drought stress, with watering carefully administered to avoid any physical contact with the leaves.

### 4.2. Collection and Preparation of Phylloplane Samples

Thirty leaves (*n* = 30; 3 leaves per plant) from *E. aureum* plants were collected at the beginning of the experiment (T0) and subsequently after three months (T3), six months (T6) and finally after twelve months (T12) of exposure in the two different indoor environments. Leaves were harvested at shoulder height from the plants using sterile forceps and placed into sterile tubes (three leaves per tube) containing phosphate buffer (50 mM Na_2_HPO_4_·7H_2_O, 50 mM NaH_2_PO_4_·H_2_O, 0.8 mM Tween-80, pH 7.0). Microbial cells were detached from the leaf surface following the protocol outlined by Stevens et al. (2021) [24], involving sonication (100 W, 42 kHz, 3 min) followed by shaking on an orbital shaker (240 rpm, 30 min). Subsequently, the resulting leaf wash suspensions were centrifuged (3000× *g*, 15 min), and the resuspended pellets were promptly stored at −20 °C until DNA isolation.

### 4.3. Gravimetric Quantification of PM Deposited on Leaves

For gravimetric quantification, Erlenmeyer flasks were filled with 50 mL of ultrapure water and one leaf was added to each one of them. The leaf was agitated in an orbital shaker for 60 min at 270 rpm, following the procedure described by Imperato et al. (2019) [18]. Subsequently, PM fractions were separated using Type 91 Whatman ashless filters with 10 µm retention and Type 42 filters with 2.5 µm retention. Prior to use, the filters were dried overnight in an oven at 60 °C, and their weights were recorded to correct for air humidity. After filtration, the filters were dried and weighed using a PIONEER precision balance (OHAUS, Lindavista, Mexico). The leaf surface area was determined using the ImageJ2 Analysis System [37], facilitating the expression of PM amount as mg·cm^−2^ of leaf area. Statistical analysis between sites for each sampling month was conducted using R (version 4.2.2). The Shapiro-Wilk test was applied to assess normality within each group. Based on the outcome, either a parametric *t*-test or a non-parametric Mann-Whitney U test was used, with Benjamini–Hochberg correction [38].

### 4.4. Black Carbon Detection in Leaf Wash Suspensions

The detection of black carbon (BC) particles within leaf wash suspensions (*n* = 20) was accomplished through a specific and highly sensitive method based on non-incandescence-related white light (WL) generation of the particles under femtosecond-pulsed illumination, as delineated by Bové et al. (2016) [39]. Before measurement, the leaf wash suspensions were defrosted from −20 °C to room temperature (24 °C). Z-stacks of the samples were captured utilizing a Zeiss LSM 880 confocal microscope (Carl Zeiss AG, Baden-Württemberg, Germany) equipped with a femtosecond-pulsed laser (810 nm, 150 fs, 80 MHz, Mai Tai DeepSee, Spectra–Physics, Andover, MA, USA) tuned to a central wavelength of 810 nm. Imaging was facilitated using a plan-apochromat 20× objective (0.8 NA). Two-photon-induced WL emission of BC particles was recorded in non-descanned mode post spectral separation and emission filtering using 400–410 nm and 450–650 nm bandpass filters. Each sample was aliquoted at 20 µL on a glass coverslip, and z-stacks were acquired from the bottom up to 35 µm within the droplet. The resulting z-stacks possessed an imaging volume of 425.1 × 425.1 × 35.1 µm^3^ with a pixel dwell time of 1.54 µs. Image acquisition was executed using ZEN Black 2.0 software.

To quantify the numbers of BC particles within the obtained z-stacks, MATLAB R2024a (MathWorks, Natick, MA, USA) was employed. A peak-searching algorithm was employed to count pixels surpassing a threshold value, set at 0.1% below the highest pixel intensity value of the narrow second-harmonic generation channel (405/10) and two-photon-excited autofluorescence channel (550/200). Thresholded pixels from both images were compared, and only the overlapping ones were considered BC particles. The average number of particles detected in the washing sample z-stacks was normalized to the imaging volume, and the results were expressed as the number of detected BC particles per mL. Statistical analysis of mean values between sites was conducted using R. The Shapiro-Wilk test was applied to assess the normality of the data. Based on the outcome, either a parametric *t*-test or a non-parametric Mann-Whitney test was used, with Benjamini–Hochberg correction [38].

### 4.5. Leaf Elemental Composition and Surface Particle Element Distribution

Leaf samples were analyzed for their elemental composition using ICP-OES. The samples were digested with 70% HNO_3_ in a heat block and dissolved in 5 mL of 2% HCl following the USEPA 3050B Acid Digestion of Sediments, Sludges, and Soils protocol [40] (Environmental Protection Agency [EPA] 1996a). The concentrations of elements were determined using inductively coupled plasma-atomic emission spectrometry (ICP-OES, Agilent Technologies 700 series, Santa Clara, CA, USA). Blank samples containing only HNO_3_ were included, and certified reference materials (Cabbage-BCR and Spinach Leaves 1570a) were included in each batch. The obtained recoveries fell within the ranges of 37–40% for Cd, 77–90% for Cu, 79–90% for Mn, 61–70% for Pb, and 79–91% for Zn.

For the study of particle element distribution, analysis was conducted using a Scanning Electron Microscope (Phenom™ ProX Desktop, Thermofisher, Waltham, MA, USA), operating at 15 kV. Portions of leaves were dried and affixed to a carbon substrate before analysis.

To assess the distribution of element concentration data, several normality tests were performed using R, including the Shapiro–Wilk test, Anderson–Darling test, and Kolmogorov–Smirnov test [41]. Based on these results, statistical comparisons were conducted as follows: for elements with normal distributions, an independent *t*-test was used to compare concentrations between sites, and a paired *t*-test was applied for within-site temporal comparisons. For elements with non-normal distributions, the Mann–Whitney U test was used for between-site comparisons, and the Wilcoxon signed-rank test was used for within-site comparisons over time.

### 4.6. Metabarcoding of the Bacterial Phylloplane

Leaf wash suspensions (*n* = 20) were subjected to centrifugation at 10,000× *g* for 10 min at 4 °C. Genomic DNA isolation was performed using the RNeasy PowerSoil Kit (QIAGEN, Hilden, Germany), automated for MagMAX™ (Thermo Fisher Scientific, Waltham, MA, USA). Initially, the sequencing library was prepared for bacterial phylloplane metabarcoding. The library was constructed by PCR amplification of the V3–V4 hypervariable region of the 16S rRNA gene. This amplification utilized 341F (5′-TCGTCGGCAGCGTCAGATGTGTATAAGAGACAGCCTACGGGNGGCWGCAG-3′) and 785R (5′-GTCTCGTGGGCTCGGAGATGTGTATAAGAGACAGGACTACHVGGGTATCTAATCC-3′) primers [42], containing Nextera (Illumina, San Diego, CA, USA) transposase adapters. The 25 µL PCR reaction comprised 1× reaction buffer with 1.8 mM MgCl_2_, 0.2 µM dNTP mix, 0.05 U µL^−1^ enzyme blend (FastStart High Fidelity PCR System, Roche, Basel, Switzerland), 0.2 µM of each primer, 5% DMSO, and 1 µL of DNA template. Amplification conditions were as follows: 95 °C for 2 min; 25 cycles of 95 °C for 30 s, 54 °C for 30 s, 72 °C for 1 min; with a final extension at 72 °C for 6 min. PCR products were purified using AMPure XP beads (Beckman Coulter, Brea, CA, USA), then indexed using the Nextera XT index kit (Illumina, San Diego, CA, USA). The 25 µL index PCR reaction included 1× reaction buffer with 1.8 mM MgCl_2_, 0.2 µM dNTP mix, 0.05 U/µL enzyme blend (FastStart High Fidelity PCR System, Roche, Basel, Switzerland), 0.2 µM of each index primer, and 1 µL of purified PCR product. Amplification conditions were the same as the initial PCR, with 14 cycles and a final extension at 72 °C for 6 min. After another purification step, the DNA concentration of each indexed sample was quantified using a Qubit dsDNA HS assay kit and the Qubit 2.0 fluorometer (Thermo Fisher Scientific, Waltham, MA, USA), before equimolar pooling. Amplicon size and integrity were confirmed with an Agilent 2100 Bioanalyzer system (Agilent Technologies, Santa Clara, CA, USA), followed by sequencing using a MiSeq Reagent Kit v3 on a MiSeq system (Illumina, San Diego, CA, USA). During the preparation of the sequencing libraries, a mock community composed of eight bacterial species was included to evaluate sequencing biases, errors, and other artifacts (ZymoBIOMICS Microbial Community DNA Standard, Zymo Research, Irvine, CA, USA).

### 4.7. Leaf Microbiome Taxonomy Analysis

Data analysis followed established procedures by Cangioli et al. (2022) [43]. The DADA2 pipeline (version 1.3) [44] was utilized for clustering amplicon sequence variants (ASVs). All ASV reconstruction and statistical analyses were conducted within the R environment version 4.2.2 [45]. After undergoing filtering, trimming, dereplication, merging, and chimera removal, bacterial taxonomy assignment was executed by comparing 16S rRNA ASVs against the SILVA_SSU_r138 database [46]. This was accomplished using a native implementation of the naive Bayesian classifier method for taxonomic assignment, facilitated by the “DECIPHER” R package (version 2.26.0) [47] as implementation of DADA2, SSU version 138 accessible at: http://www2.decipher.codes/Downloads.html (accessed on 11 March 2024). The “assignTaxonomy()” function was employed for this purpose. Annotated ASV count tables were further processed using the Phyloseq package [48].

Calculation of alpha diversity (Shannon and Pielou’s Evenness indices), phylogenetic diversity (Faith’s PD index), Good’s coverage (diversity coverage index), and richness ASVs index was performed using the “alpha()” function within the “microbiome” R package (version 1.20.0). To assess statistical differences between sites after a year, Wilcoxon rank-sum tests were applied using the “Stat compare means ()” function from the “ggpubr” R package (version 0.6.0), with Benjamini-Hochberg correction for multiple testing.

Taxonomic disparities among microbiota were visualized via principal coordinates analysis (PCoA), utilizing the “ordinate” function and plotted using the “plot_ordination()” function within the phyloseq package. Rarefaction curves were constructed using the “rarecurve()” function from the R packages “vegan” (version 2.6.4), applied to the phyloseq object. Relative abundance plots were generated using the “ggplot2” R package (version 3.5.0).

The compositional distribution of each sample at different taxonomic levels (phylum, class, order, family, and genus) was visualized using the “plot_bar()” function from the R package “phyloseq” (version 1.42.0), applied to a phyloseq object constructed from the OTU table, sample metadata, and taxonomic information. Prior to visualization, the top 20 most abundant genera were selected and normalized to account for differences in sequencing depth across samples.

## 5. Conclusions

This study offers a comprehensive assessment of the capacity of *E. aureum* to remove harmful airborne pollutants with a focus on its phylloplane microbial communities. Significant variations in microbial alpha diversity indices suggest that PM-Pb, particularly from gunshot residues, substantially alters the composition and diversity of these microbial communities. Temporal and site-specific differences further highlight how environmental factors shape microbial dynamics, suggesting that Pb pollution may decrease both the diversity and/or variability of the phylloplane microbiome. These findings enhance our understanding of plant-microbe interactions under pollution stress and provide a foundation for developing strategies for modeling and manipulating these highly beneficial microbial consortia for ecological and economic applications.

## Figures and Tables

**Figure 1 plants-14-02956-f001:**
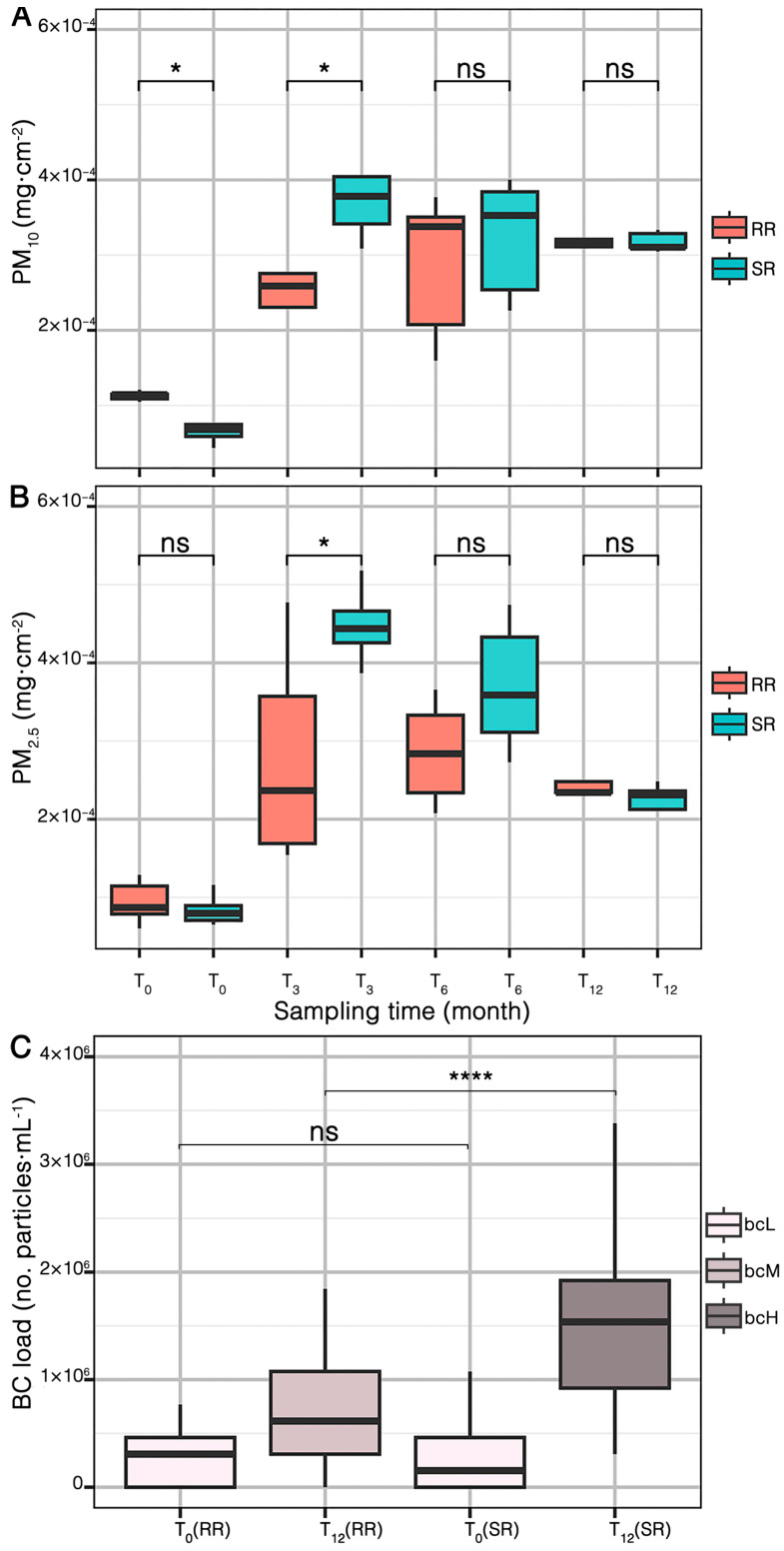
Boxplots comparing the accumulation of PM_10_ (**A**), PM_2.5_ (**B**) and Phylloplane BC load (**C**) monitored for 1 year among reference room (RR) and shooting room (SR). Box plots span the interquartile range (25th to 75th percentile), lines within boxes denote the median and black lines extend to 1.5 times the interquartile range. Low (bcL), medium (bcM), and high (bcH) phylloplane BC load. *, **** and ns refers to *p* values <0.05, <0.0001 and >0.05, respectively.

**Figure 2 plants-14-02956-f002:**
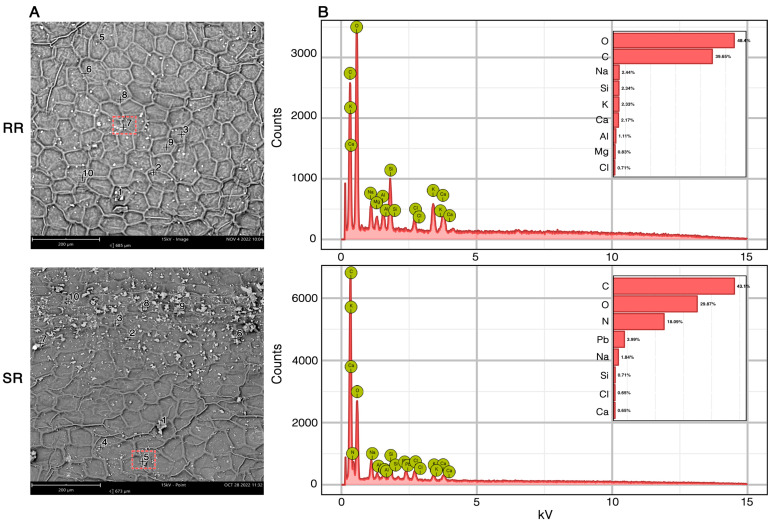
SEM images (**A**) of *E. aureum* leaves after 12 months of exposure in the reference room (RR) and the shooting room (SR). Numbers in black followed by a cross represent the specific locations where EDX spectra were taken. The spectrum with element weight percentages present at RR and SR (**B**) corresponds to the locations highlighted with red dashed frame (location 7 for RR and location 5 for SR).

**Figure 3 plants-14-02956-f003:**
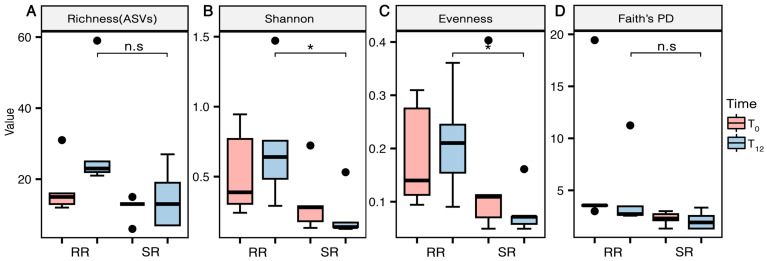
Comparison of alpha diversity and phylogenetic indices of phylloplane microbial communities between reference (RR) and shooting (SR) rooms at initial time (T0) and after 12 months (T12). The indices include Observed Species (**A**), Shannon Diversity (**B**), Pielou’s evenness (**C**) and Faith’s Phylogenetic Diversity (**D**). * and ns refers to *p* values <0.05 and >0.05, respectively.

**Figure 4 plants-14-02956-f004:**
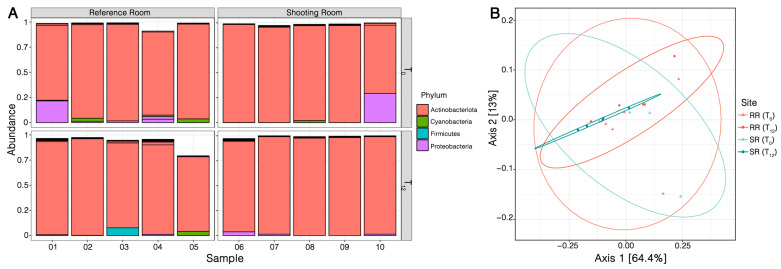
Relative abundances of taxa at the phylum level (**A**) and Principal Coordinates Analysis (PCoA) (**B**) of phylloplane samples from *E. aureum* leaves located in the reference room (RR) and the shooting room (SR) at the initial time (T0) and after 12 months (T12).

**Figure 5 plants-14-02956-f005:**
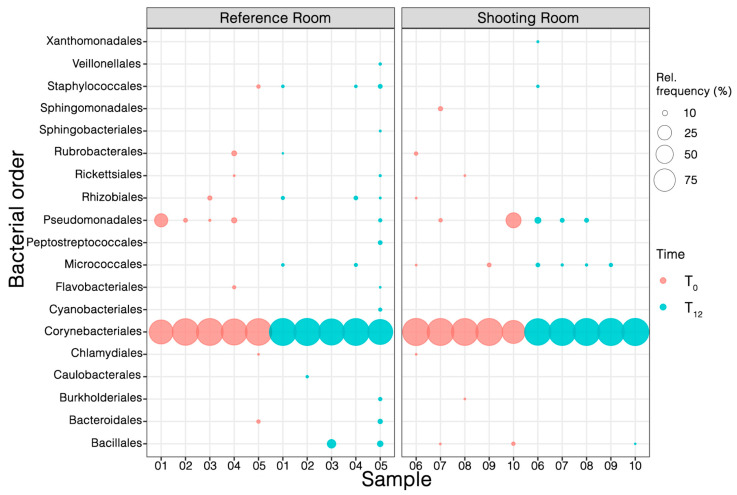
Bubble plot depicting the relative frequency (as a percentage) of the bacterial Orders across phylloplane samples. Only bubbles with ≥2% relative abundance are shown. Samples are grouped by sites (Reference room or Shooting room) and bubbles are colored by sampling time (month).

**Table 1 plants-14-02956-t001:** Summary statistics of elemental concentrations (mg kg^−1^ dry weight) in *E. aureum* leaves from the reference room (RR) and shooting room (SR). Results are expressed as the mean of five replicates ± SD. Element concentrations were determined by ICP-OES.

	RR				SR			
	T_0_	T_3_	T_6_	T_12_	T_0_	T_3_	T_6_	T_12_
Ca	212.6 ± 84.3	192.1 ± 19.8	491.0 ± 294.3	219.2 ± 50.3 ^S^	240.9 ± 60.4	354.5 ± 144.4	333.2 ± 138.7	472.1 ± 152.6 ^S^
Cd	0.01 ± 0.01 ^S^	0.02 ± 0.002	n.d. ^1^	0.01 ± 0.003 ^S^	n.d. ^S^	0.02 ± 0.003	n.d.	0.02 ± 0.003 ^S^
Cu	0.08 ± 0.046	0.09 ± 0.075	0.25 ± 0.15	0.16 ± 0.05	0.09 ± 0.03	0.18 ± 0.148	0.19 ± 0.10	0.16 ± 0.05
Fe	1.63 ± 0.25	4.92 ± 2.27	4.59 ± 2.34	5.23 ± 0.42	1.59 ± 0.49	3.75 ± 1.68	3.88 ± 2.80	4.34 ± 1.39
K	188.7 ± 47.7	198.0 ± 50.9	227.0 ± 13.6	222.7 ± 38.3	199.6 ± 26.7	225.7 ± 66.5	173.4 ± 59.4	228.7 ± 46.0
Mg	34.4 ± 11.5	31.3 ± 5.76 ^S^	34.2 ± 10.6	41.0 ± 3.58 ^S^	38.8 ± 9.01	48.5 ± 12.0 ^S^	36.6 ± 14.1	67.9 ± 10.1 ^MS^
Mn	6.91 ± 5.33	6.86 ± 2.75	7.71 ± 3.18	17.3 ± 2.88 ^M^	8.18 ± 3.93	13.5 ± 6.81	9.29 ± 2.95	16.3 ± 5.18
Na	74.6 ± 34.1	87.4 ± 20.1	98.6 ± 40.9	158.8 ± 16.4	88.8 ± 33.1	104.2 ± 33.0	83.6 ± 32.9	136.5 ± 26.9
P	28.8 ± 14.0	45.0 ± 11.2 ^S^	44.2 ± 6.6	59.5 ± 15.7	34.9 ± 7.4	72.0 ± 11.1 ^MS^	51.8 ± 20.7	73.5 ± 11.0
S	13.9 ± 4.60	17.5 ± 3.30	18.6 ± 2.44	22.6 ± 3.90	16.1 ± 2.60	22.2 ± 7.34	16.5 ± 6.47	28.2 ± 5.92
Zn	0.68 ± 0.35	1.06 ± 0.31	0.87 ± 0.40	1.23 ± 0.31 ^S^	0.98 ± 0.34	1.45 ± 0.64	1.02 ± 0.49	1.87 ± 0.71 ^MS^

^1^ n.d. = not detected. ^M^ denotes significantly different (*p* < 0.05) in element concentration compared to the previous sampling time within the same site. ^S^ denotes significantly different (*p* < 0.05) in element concentration between sites at the same time point.

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
