# Peer review of "Evaluating the Capability of *Epipremnum aureum* and Its Associated Phylloplane Microbiome to Capture Indoor Particulate Matter Bound Lead"

_plants, 2025, doi:10.3390/plants14192956_

Round 1

Reviewer 1 Report

Comments and Suggestions for Authors
The most novel and compelling aspect of this study is the comparison of plant performance between a standard office environment and a firearm shooting range.
I recommend that the authors revise the manuscript to emphasize this unique aspect more prominently in the title, abstract, and introduction.
Additionally, the introduction should contextualize the study by discussing air pollution and PM exposure specifically associated with shooting firearms.
The intro and discussion can be expanded to include more comparison with leaf PM accumulation research, especially as only pothos was studied in the current work.
The description of the plants locations within each room needs improvement. Possible include a blueprint of the rooms to indicate their location.
Both 5 and Figure 6 are rather redundant in the visual communication, and they are presenting the same information over and over. I think these figures can be moved to the Supp, and referred to as required.  

Author Response

Comments 1: The most novel and compelling aspect of this study is the comparison of plant performance between a standard office environment and a firearm shooting range.

I recommend that the authors revise the manuscript to emphasize this unique aspect more prominently in the title, abstract, and introduction.

Response 1: Thank you for pointing this out. We have emphasized this in lines 27-29 and 80-83.

Comments 2: Additionally, the introduction should contextualize the study by discussing air pollution and PM exposure specifically associated with shooting firearms.

Response 2: We have updated the introduction to include a discussion on air pollution and PM exposure specifically associated with shooting firearms. This addition, can be seen in lines 41–48 and 50-58 of the revised manuscript.

Comments 3: The intro and discussion can be expanded to include more comparison with leaf PM accumulation research, especially as only pothos was studied in the current work.

Response 3: Thank you for your suggestion. We have expanded the introduction to include a comparison with previous research on leaf PM accumulation in multiple plant species, highlighting studies on Ipomoea batatas L., Hedera helix L., and Epipremnum aureum. These additions can be seen in lines 63–65 of the revised manuscript.

Comments 4: The description of the plants locations within each room needs improvement. Possible include a blueprint of the rooms to indicate their location.

Response 4: We agree with this comment. Therefore, we have added the Figure S1: Floor plan of the reference rooms (RR) and shooting room (SR) housing the plants, withing Supplementary Materials section.

Comments 5: Both 5 and Figure 6 are rather redundant in the visual communication, and they are presenting the same information over and over. I think these figures can be moved to the Supp, and referred to as required. 

Response 5: We have moved Figure 6 to the Supplementary Material as suggested. However, we decided to retain Figure 5 in the main text because it provides a clear visual representation that helps interpret the data presented in lines 191–199.

Reviewer 2 Report

Comments and Suggestions for Authors

This study evaluated over a 1-year period, the ability of Epipremnum aureum leaves to collect particulate matter (PM)-bound Pb from an indoor environment. Using Illumina MiSeq, this study investigated the changes in the phylloplane microbiome connected with the accumulation of this pollutant. Plants were placed in a shooting room, where PM release from each shot was recorded, along with PM2.5 and PM10 sequestration and leaf element enrichment by ICP. Additionally, black carbon (BC) sequestration was determined, and SEM-EDX was performed on leaves after 12 months of exposure. Our results indicated that ambient air pollution shapes microbial leaf communities by affecting their diversity. At the order level, Pseudomonadales, along with Micrococcales, appeared (at a low relative abundance) after exposure to indoor PM-bound Pb air pollution. The Epipremnum aureum air filtration approach holds promise for reducing indoor air pollution, but more knowledge about the underlying mechanisms supporting genera capable of coping with airborne pollutants is still required. Therefore, I suggest that this paper be accepted with further modification.

1Line 66 PM2.5 and PM10 should be subscripted

2Line345 Please verify if it is correct

TCGTCGGCAGCGTCAGATGTGTATAAGAGACAGCCTACGGGNGGCWGCAG-3

3Line347 Please verify if it is correct

TCTCGTGGGCTCGGAGATGTGTATAAGAGACAGGACTACHVGGGTATCTAATC347 C-

4There should be some new references.

Benka-Coker, W.; Sipe, K.; Dedic, D.; Jones, A.; Hawkins, B.; Lyons, E.; Steiman, M.; Benka-Coker, M. Exploring the Relative Effects of Natural Gas and Biogas Cooking on Indoor Air Quality in Residential Kitchens. Atmosphere 2025, 16, 1061. https://doi.org/10.3390/atmos16091061

 A Hybrid Wavelet-Based Deep Learning Model for Accurate Prediction of Daily Surface PM2.5 Concentrations in Guangzhou City. Toxics. 2025; 13(4):254. https://doi.org/10.3390/toxics13040254

Vega, E.; Wellens, A.; Namdeo, A.; Meza-Figueroa, D.; Ornelas, O.; Entwistle, J.; Bramwell, L. Spatial Variation of PM10 and PM2.5 in Residential Indoor Environments in Municipalities Across Mexico City. Atmosphere 2025, 16, 1039. https://doi.org/10.3390/atmos16091039

  1. Through further proofreading, the quality of this article should be greatly improved.

Author Response

Comments 1: Line 66 PM2.5 and PM10 should be subscripted

Response 1: We have, accordingly, done it

Comments 2: Line345 Please verify if it is correct

TCGTCGGCAGCGTCAGATGTGTATAAGAGACAGCCTACGGGNGGCWGCAG-3

Response 2: We have verified the sequence at line 345, and the code is correct.

Comments 3: Line347 Please verify if it is correct

TCTCGTGGGCTCGGAGATGTGTATAAGAGACAGGACTACHVGGGTATCTAATC347 C-

Response 3: We have verified the sequence at line 347, and the code is correct.

Comments 4: There should be some new references.

Benka-Coker, W.; Sipe, K.; Dedic, D.; Jones, A.; Hawkins, B.; Lyons, E.; Steiman, M.; Benka-Coker, M. Exploring the Relative Effects of Natural Gas and Biogas Cooking on Indoor Air Quality in Residential Kitchens. Atmosphere 2025, 16, 1061. https://doi.org/10.3390/atmos16091061

A Hybrid Wavelet-Based Deep Learning Model for Accurate Prediction of Daily Surface PM2.5 Concentrations in Guangzhou City. Toxics. 2025; 13(4):254. https://doi.org/10.3390/toxics13040254

Vega, E.; Wellens, A.; Namdeo, A.; Meza-Figueroa, D.; Ornelas, O.; Entwistle, J.; Bramwell, L. Spatial Variation of PM10 and PM2.5 in Residential Indoor Environments in Municipalities Across Mexico City. Atmosphere 2025, 16, 1039. https://doi.org/10.3390/atmos16091039

Response 4: We have added the references suggested by the reviewer in line 41-46.